# A Cost Analysis of an Outreach School-Based Dental Program: Teeth on Wheels

**DOI:** 10.3390/children8020154

**Published:** 2021-02-18

**Authors:** Tan Minh Nguyen, Utsana Tonmukayakul, Hanny Calache

**Affiliations:** 1Deakin Health Economics, Institute of Health Transformation, Faculty of Health, Deakin University, Waurn Ponds, VIC 3216, Australia; utsana.tonmukayakul@deakin.edu.au (U.T.); hanny.calache@deakin.edu.au (H.C.); 2Community Dental Program, Peninsula Health, Frankston, VIC 3199, Australia; 3Coburg Hill Oral Care, Coburg North, VIC 3058, Australia; 4Dentistry and Oral Health, La Trobe Rural Health School, La Trobe University, Bendigo, VIC 3552, Australia

**Keywords:** health promotion, dental care for children, school health services, oral health, costs and cost analysis

## Abstract

Background: This study evaluated an outreach mobile dental service called Teeth on Wheels (TOW). The dental program targeted Australian children from low household income, who are eligible for the Child Dental Benefits Scheme (CDBS) in Victoria, Australia. The program is complemented with a school-based oral health promotion element. Methods: A retrospective cohort study was performed with a convenience sample. Children must have had at least three dental examinations during the 2016–2019 calendar years to be included in the study. Comparisons were made between the 2016–17 and 2018–19 calendar years. It was hypothesised that the program would result in reduced costs and the number of restorations and extractions in the latter period. Results: A total of 414 children were included in the analysis. The total mean costs of the program per child reduced from AU$605.3 in 2016–17 to AU$531.1 in 2018–19. The results showed an overall mean reduction in all restorations and extractions performed, but only statistical significance was noted for reductions of restored deciduous teeth. Conclusions: This outreach program, which is focused on prevention and minimally invasive dentistry, can be a promising alternative model of delivery for dental services in young children.

## 1. Introduction

School-based dental services (SDS), delivered via mobile dental vans or fixed dental clinics near schools, have traditionally been the model of delivery of public dental care for children in Australia. However, community needs, expectations, and government policy directives have changed over time and have led to a different model of dental service delivery for children, and they vary between Australian state and territory jurisdictions. SDS have been the cornerstone of many government-funded dental programs internationally, targeting unmet oral health needs among children, and its workforce has primarily consisted of dental therapists [1,2].

Some Australian states like Victoria and New South Wales have ceased SDS and integrated child and adult dental services within centralised community health services [3]. Although many countries have limited dental therapists to practice only in government-funded dental programs [1,2], recent regulatory changes to their scope of practice standard have enabled independent practice in Australia [4], and they are permitted to work in both the public and private sectors. In recent times, there has been a renewed emphasis on giving user choice through consumer-directed care for publicly funded dental services [5]. i.e., users of Australian public dental care should be given more flexibility in seeking and receiving services from a dental provider of their choice.

Interventions that reduce childhood oral health inequities remain elusive, largely because dental caries, the most common oral disease among children, has multifactorial influences. These interventions are emphasised under three broad themes underpinned by the Fisher-Owens’ model: (1) child-level influences, (2) family-level influences, and (3) community-level influences [6]. School-based dental programs can be implemented in many different forms, including focusing only on oral health promotion initiatives without necessarily involving the provision of dental services. For example, a school-based toothbrushing program with fluoride toothpaste have been demonstrated to significantly reduce dental caries experience [7,8].

The school environment is an ideal setting for targeted interventions that promote positive oral health behaviours and address child oral health inequities. Nevertheless, there is anecdotal reports that oral health promotion in schools are rarely implemented in Victoria, possibly since these activities are not funded by the Victorian public sector [9]. Furthermore, how to best increase the attendance rate of children from low income household to receive timely dental care remains a key challenge. Previous systematic reviews have found that school-based dental screening has had limited effectiveness to increase the utilisation of dental services by children [10,11]. The provision of mobile dental services on-site at school premises appears to be the most effective approach to increase dental attendance by children from low income households [12].

A specific focus to tackle child oral health inequities by the Australian government emerged with the introduction of the Child Dental Benefits Scheme (CDBS) on 1 January 2014. The CDBS is a means-tested federally funded dental program that provides eligible children (aged 2–17 years) from low household income, up to $1000 capped value of dental care over a two calendar-year period. Consistent with the broad objective to increase choice on consumer-directed public dental care, children can access dental services through the CDBS in either the public or private sector.

Although there is no out-of-pocket expenses to the children’s families (unless their capped benefit is reached) the utilisation rates of the CDBS has been low, starting from 29.5% in 2014 to 37.9% in 2018 [13]. Just over one-third (40.5%) of children aged 5–6 years from low household income have never visited a dental practitioner [14]. It is plausible that the low uptake of dental services under the CDBS may be associated with current models of dental care delivery in Australia. This is predominantly delivered through fixed dental clinics, in private practice, which are maldistributed geographically [15], and largely provided by dentists (72% of the oral healthcare workforce) [16].

Inspired by the purpose to advocate for equity in oral health, the Teeth on Wheels (TOW) dental program was established in 2015. It is a privately run outreach mobile dental service operating in Victoria and New South Wales and is primarily focused on working with primary schools [17]. TOW currently has a database of over 50,000 children, partnered with over 400 educational facilities and has a specific focus on prevention and minimally invasive dentistry. The outreach dental program predominantly provides services to CDBS eligible children.

Children are typically seen twice a year for regular dental examinations and preventive treatments such as scale and clean or oral prophylaxis, topical fluoride applications (fluoride varnish) and fissure sealants. Other dental services provided include restorations and extractions, and where appropriate, a referral to specialist paediatric care. All dental services are provided using formal informed consent processes with the parents and in co-ordination with the educational facilities.

Additionally, TOW has integrated, as part of their service, an oral health education element that is delivered collaboratively with the school staff. This includes in classroom oral health education directly with teachers and students. Also included in this oral health education program is full school assembly education sessions to introduce and build relationships between the dental team with the whole school community including school staff, the children, and their parents.

The TOW dental program is facilitated by separate teams who manage different aspects of the business operations. One team works with schools on administrative tasks required for the implementation of their mobile dental service and oral health promotion activities. This allows schools to maintain their normal day-to-day activities without any disruptions. The TOW program can be modified where appropriate to accommodate any specific requirement of schools such as the delivery of services to children with additional needs or are medically compromised.

Another team works with the parents in the provision of oral health education advice and addressing any specific questions that the parents may have about the TOW mobile dental service or the oral health and dental care of their child. Advice is offered for parent-focused communication to relieve dental anxiety and to support the uptake of timely dental treatment. The TOW team organises follow-up visits with the parents to ensure that the children at high-risk to oral diseases have ongoing preventive-focused dental care. Parents were not necessarily required to attend their child’s dental appointment, but the option is offered.

The TOW clinical team are trained to communicate with children in an appropriate manner by building trust and ensuring that the experience is fun and memorable, even when dental treatment may not be possible due to child co-operation barriers. i.e., this approach is critical for the achievement of “The Positive Dental Experience” that the TOW dental program promises. The mobile dental vans are designed to create a relaxing and calm environment, such as movie-themed adventures. Educational toys are given as rewards once dental treatment is completed, and children can participate in educational games such as choosing healthy and unhealthy foods on a wall board. Positive language is used by the clinical team to accommodate difficult scenarios. For example, a filling is known as a ‘star’ and the high-speed handpiece is known as the ‘water toothbrush’.

The aim of this study is to perform a retrospective cost-analysis of the TOW program between the 2016–17 calendar years and 2018–19 calendar years. It is hypothesised that the cost and number of dental treatment services provided by TOW in the 2016–2017 years will be greater than that provided to the same cohort of children in 2018–19. A secondary hypothesis is that the cost and number of preventive services are likely to be similar between the two time periods since they are essential oral healthcare services that is intended to promote and maintain oral health.

## 2. Materials and Methods

### 2.1. Data Collection

A retrospective convenience sample of children living in Victoria was used following a quality assurance audit of dental records. Primary data was supplied by TOW to the principal investigator (TMN), which included the diagnosis of tooth-level dental caries and dental treatment services of a cohort of children who received dental care between January 2016 to December 2019.

A minimum sample size estimation of 140 was derived from a previous study of Australian children receiving a minimum intervention dentistry approach to dental caries or standard care [18]. (Alpha = 0.05 one-sided, power = 0.90, m1 = 4.4 ds, m2 = 8.6 ds, sd1 = 9.3, sd2 = 11.0, *n*2/*n*1 = 1.0; m = mean, ds = decayed surfaces, sd = standard deviation, *n* = population size).

The eligibility criteria to be included in this study required children having one dental examination in 2016 (baseline), one dental examination in 2019 (follow-up), and at least one dental examination in 2017 or 2018, i.e., children included in the study were required to have at least three dental examinations during the four-year study period, 2016 to 2019.

Since the data used in this study is data collected during service delivery of a dental program, the oral health examiners who collected the data were uncalibrated. In practice as per personal communications with the company directors [17], the dental caries diagnosis is typically made according to the International Caries Detection and Assessment System II classification [19] with at least cavitated enamel caries level (d_3_/D_3_).

Any teeth that were diagnosed with dental caries, restored or missing (extracted) after the baseline dental examination on the child’s odontogram was calculated as the cumulative carious teeth for the deciduous or permanent teeth (tooth-level). Any child with at least one new carious tooth in either dentition was designated to experience cumulative caries incidence (any dentition, person-level). Annual caries incidence and annual carious teeth were not possible to be determined due to missing data. Due to the lack of data on the reasons for the restoration of permanent anterior teeth (which could be due to dental caries or trauma) and the fact that posterior permanent teeth are at a much greater risk of developing dental caries than anterior permanent teeth, it was decided to exclude restored anterior permanent teeth as a count for dental caries.

The costs of and the number of dental treatment services were classified using the CDBS item code schedule [20], consistent with the descriptions of the Australian Schedule of Dental Services and Glossary [21]. For fissure sealants, restoration and extraction procedures, the dentition, on which these procedures were performed, was noted—i.e., whether the procedure was undertaken on a deciduous or permanent tooth.

This research received an ethics exemption from the Deakin University Human Research Ethics Committee (ID 2020-100). The details of the study consistent with the STROBE checklist is provided as a Appendix A.

### 2.2. Data Analysis

Data were cleaned using Excel 365 (Microsoft Corporation™, Washington, DC, USA). Summary descriptive statistics were reported according to the age, gender, and the child’s principal place of residence according to the Socio-Economic Indexes for Areas (SEIFA) [22], and the Australian Statistical Geography Standard—Remoteness Area classification (ASGC-RA) [23]. The SEIFA ranks areas in Australia according to relative socio-economic disadvantage and advantage ranging from 1–10.

The total costs in 2019 from a healthcare perspective and relevant one-group mean-comparison tests (*t*-tests) were performed using Stata IC Version 12 (Statacorp™, College Station, TX, USA), where statistical significance was determined at *p* < 0.05.

The cost-analysis was grouped according to preventive services (includes dental check-up, intraoral radiographs, prophylaxis, scaling, topical fluoride applications and fissure sealants), treatment services (restorations and extractions) for deciduous teeth or permanent teeth, and the total costs for preventive and treatment services (combined). The costs, and the number of dental services per 100 individuals were calculated and compared between the 2016–17 years and 2018–19 years.

## 3. Results

A total of 414 children met the inclusion criteria in the study, with the summary statistics reported in Table 1. About half (52%) of children were male. The mean age was 6.93 (1.82 SD) and 9.90 (1.83 SD) when they received a dental examination at baseline and at follow-up, respectively.

Just over half (52%) of children lived in a geographic area in the lowest 50th percentile for socioeconomic disadvantage according to the SEIFA classification, 47% was living in the highest 50th percentile, and 1% was living where the SEIFA classification is unknown.

Just over half (55%) of children were living in the RA2 classification of “Inner Regional Australia”, 45% were in the RA1 classification of “Major Cities of Australia” and <1% was living in an area where the ASGC-RA classification is unknown.

The mean follow-up period between the baseline examination in 2016 and the final follow-up examination in 2019 was 2.96 years (95% CI 2.91; 3.02).

The cumulative caries incidence (any dentition) is 0.15 (SD 0.36), cumulative carious deciduous teeth is 0.17 (SD 0.63) and cumulative carious permanent teeth is 0.12 (SD 0.54). When converted, the annual caries incidence and the rate of annual carious deciduous and permanent teeth were 0.05, 0.06, and 0.04, respectively.

The cost-analysis is reported in Table 2. There were statistically significant differences noted between the two time periods for the costs of treatment services for deciduous and permanent teeth, and the total costs for preventive and treatment services (combined). There was no statistically significant difference noted when the costs of preventive services were analysed across the two time periods.

The rate of dental services per 100 individuals is reported in Table 2. There were statistically significant differences noted for most of the different types of services provided, except for restorations performed on permanent teeth and extractions performed on deciduous teeth (no extractions were performed on permanent teeth).

There was less services provided for intra-oral radiographs, scale and clean, fissure sealants applied for deciduous teeth, and restorations performed on deciduous and permanent teeth, and extractions performed on deciduous teeth in the 2018–2019 years compared to 2016–2017 years. Conversely, there were greater services provided for examinations, prophylaxis, topical fluoride applications, and fissure sealants applied for permanent teeth in the 2018–2019 years.

## 4. Discussion

This study demonstrated that children in the target population utilising the TOW dental program, had a steady decline in the costs and dental treatment services for restorations performed on deciduous and permanent teeth, and extractions performed on deciduous teeth. The lower costs and the lower rate of dental treatment services in the 2018–19 years compared to the 2016–17 years was consistent with the a priori hypothesis. In contrast, the mean differences for the costs associated for preventive dental services were not statistically significant, although different types of preventive services were highly variable, indicating the program had a continuous strong focus on prevention.

Although the results did not specifically show that the TOW dental program increased the utilisation of dental services by children in comparison to ‘no intervention’ or ‘standard care’ (users of any dental services), it illustrates public value by complementing the outreach mobile dental service with school-based oral health promotion activities. Anecdotal feedback by the school community has been overwhelmingly appreciative with respect to creating positive dental experiences for children. There are also societal costs benefits of the program not captured in this study, such as the potential opportunity costs lost that would have occurred if the parents were required to arrange work leave arrangements to take their child to a fixed dental service.

From a health economics and equity perspective, it is still unknown how to best address child oral health inequities using school-based settings. The TOW program appears to have reached a higher proportion of children whose principle place of residence is in the lowest 50th percentile on the SEIFA index (47%) compared to children receiving standard public dental care based on a previously published study (29%) [24]. A literature review of SDS has shown that they reduce the prevalence of untreated dental caries [2], but their level of effectiveness to reduce dental caries incidence remain unclear. Alternative models of care, which exclude treatment services, such as school-based dental screening, are reported to be common practice in Australia, perhaps even globally, despite strong evidence demonstrating limited clinical benefits [10,11].

The annual caries incidence and the rate of annual carious permanent teeth in this study are lower than those reported in the systematic review and meta-analysis study, reporting 0.11 and 0.18, respectively [25]. However, these observations are principally based on epidemiological studies inclusive of a child and adult population, rather than longitudinal studies of the same child population cohort. The evaluated dental caries outcomes from the TOW program can help inform the economic modelling and budget impact analysis for future program expansion from an Australian perspective.

In Victoria, SDS ceased in 2007 and merged into community health services [24]. Advocates for community health services integrating dental care have argued that it promotes family-centred multi-disciplined co-ordinated healthcare. However, there are societal trade-offs where there is a reliance for consumer-driven demand, particularly for families from lower socioeconomic backgrounds. National statistics report a greater proportion of children from low household income have never visited a dental practitioner compared to middle and high household income, with similar patterns evident for the prevalence of untreated decayed teeth [14].

Previous research in Victoria has shown that an outreach school-based dental check-up program can address childhood oral health inequities by increasing the utilisation of public dental services, had a high child retention for follow-up dental care via referral, and was less costly compared to ‘standard care’ (users of fixed dental clinics) [26,27]. Other models of care with high potential are school-based interventions that includes one or more preventive services such as fluoride varnish, silver fluoride, fissure sealants, and interim atraumatic restorations. For example, a comprehensive caries prevention program called ‘CariedAway’ in the United States was shown to be highly cost-effective [28]. Logically, there is no ‘one size fits all approach’ to address childhood oral health inequities, and different public health interventions may be necessary depending on the target population context.

Furthermore, more effort is needed to reduce oral health inequities for Indigenous Australians [29,30]. National statistics indicate that the differences in the prevalence of untreated dental caries among Indigenous Australian children are profound, at almost a two-fold burden compared to their non-Indigenous counterparts [14]. There seems to be promise that the CDBS is closing some levels of childhood oral health inequities. Aboriginal and non-Aboriginal children had similar levels of utilisation between the four financial years from 2013–14 to 2016–17, but did have lower rates of preventive services [31].

To date, few studies have evaluated the longitudinal utilisation of the CDBS by Australian children nor the impact of dental services to reduce caries incidence. In the 2014–15 financial year, 45.6% of the total cost expenditure under the CDBS was for diagnostic and preventive services [32]. This proportion changed to 59.7% for the 2014–15 calendar years [33]. The contrasting proportions could be partly explained by differences in data time periods, and perhaps children in the later years are less likely to require dental treatment services.

A shift towards a higher proportion of costs on preventive services in later years was observed in our study. However, these proportions appear significantly higher at 84.7% and 93.8% for the 2016–17 and 2018–19 calendar years, respectively. These values provide evidence the TOW program had a strong focus on prevention and minimally invasive dentistry. Future research would be beneficial to determine the clinical effectiveness of children receiving dental care by the TOW program using more robust study methods to validate our study findings, particularly, the utilisation of calibrated dental practitioners to diagnose caries experience.

### Limitations

This study did not employ an experimental design which would have given a more robust comparative analysis. The selected research approach was restricted by the nature of the data source. Data used in this study was primarily collected for routine practice and internal auditing purpose. Hence, only a comparison between two time periods could be conducted. Thus, the results of the study should be interpreted with caution with considerations for the following limitations.

Firstly, the cumulative dental caries incidence, carious deciduous teeth and permanent teeth were dependent on uncalibrated dental practitioners making the diagnosis and performing dental treatment. Therefore, the true results may be over- or underestimated. We believe it is likely to be overestimated since a previous literature review has found that restorative treatment is more likely to be performed when more minimally invasive treatment options are indicated [34].

Secondly, data was only confined to dental treatment provided by the TOW dental program to children that met the study inclusion criteria. The exclusion of other children from this study, to ensure the data could be interpreted appropriately, meant that the cost-analysis and dental caries outcomes would be biased. Furthermore, any dental treatment that were provided externally to the TOW dental program was not incorporated in the results. This is particularly relevant where urgent dental treatment was necessary and sought elsewhere, or if children required specialist paediatric dentistry services that require general anaesthesia.

Thirdly, it is difficult to determine: (1) whether the program increased regular utilisation of dental services relative to the child’s oral disease risk, and (2) the clinical effectiveness that may be associated with the school-based oral health promotion element provided alongside clinical dental care. Participating schools have reported positive feedback as many families were unaware their child was eligible for the CDBS benefits. Therefore, the TOW dental program is a promising alternative to the publicly funded model of care and has potential to address child oral health inequities in the target population.

## 5. Conclusions

This study showed that a sample of Australian children receiving school-based oral health promotion delivered alongside an outreach mobile dental service, which is focused on prevention and minimally invasive dentistry, had a steady decline in the number of restorative and extraction services resulting in lower costs in the 2018–19 years compared to the 2016–17 years. The TOW program delivered a high proportion of costs for diagnostic and preventive services, as would be expected under best practice according to a minimally invasive dentistry approach.

## Figures and Tables

**Table 1 children-08-00154-t001:** Summary of the descriptive statistics of children receiving dental care from the Teeth on Wheels dental program

Demographics and Dental Caries Experience	2016–17 Years	2018–19 Years
	*n* (%)	
Mean Age (SD) (years)	6.93 (1.82)	9.90 (1.83)
Gender		
Female	200 (48)	
Male	214 (52)	
Principal Place of Residence (SEIFA Classification)		
Unknown	3 (1)	
1–5	197 (47)	
6–10	214 (52)	
Remoteness Principle Place of Residence (ASGS-RA Classification)		
Unknown	1 (<1)	
RA2—Inner Regional Australia	227 (55)	
RA1—Major Cities of Australia	186 (45)	
Cumulative Caries Incidence (SD)	0.15 (0.36)	
Cumulative Carious Deciduous Teeth (SD) (*dt*)	0.17 (0.63)	
Cumulative Carious Permanent Teeth (SD) (*DT*)	0.12 (0.54)	

SD = standard deviation; SEIFA = Socio-Economic Index For Areas; 1–5 = Lowest 50th percentile score; 6–10 = Highest 50th percentile SEIFA score; ASGC-RA = Australian Statistical Geography Standard—Remoteness Area; *dt* = decayed teeth in the deciduous dentition; *DT* = decayed teeth in the permanent dentition.

**Table 2 children-08-00154-t002:** Statistical analysis of the mean costs and dental treatment services utilised by children via the Teeth on Wheels dental program

	2016–17 Years	2018–19 Years	Mean Difference (SE)	*p*-Value
**Healthcare Cost Categories**				
(A) Costs for Preventive Services (SD)	AU$512.9 (221.2)	AU$498.6 (213.7)	−AU$14.3 (14.6)	0.327
(B) Costs for Treatment Services—Deciduous Teeth (SD)	AU$77.5 (152)	AU$24.7 (83.1)	−AU$52.8 (7.5)	<0.001 *
(C) Costs for Treatment Services—Permanent Teeth (SD)	AU$14.9 (78.9)	AU$6.9 (31.9)	-AU$8.1 (4.0)	0.046 *
Total Costs for Preventive and Treatment Services (SD) (A + B + C)	AU$605.3 (287.2)	AU$531.06 (233.0)	−AU$74.26 (16.7)	<0.001 *
**Dental Services Provided (per 100 individuals)**				
Examinations (SE)	213.0 (4.0)	290.3 (4.4)	77.3 (6.5)	<0.001 *
Intra-oral Radiographs (SE)	139.6 (7.5)	67.4 (4.7)	−72.2 (8.7)	<0.001 *
Prophylaxis (SE)	24.4 (2.1)	164.5 (5.5)	140.1 (5.8)	<0.001 *
Scale & Clean (SE)	178.7 (3.7)	112.3 (4.9)	−66.4 (5.9)	<0.001 *
Topical Fluoride Applications (SE)	187.7 (4.1)	238.2 (5.6)	50.5 (6.5)	<0.001 *
Fissure Sealant Applied for Deciduous Teeth (SE)	78.0 (7.8)	8.0 (2.5)	−70.0 (8.1)	<0.001 *
Fissure Sealant Applied for Permanent Teeth (SE)	2.7 (1.5)	175.8(11.4)	173.2 (11.6)	<0.001 *
Restorations Performed on Deciduous Teeth (SE)	51.0 (5.1)	15.9 (2.8)	−35.0 (5.3)	<0.001 *
Restorations Performed on Permanent Teeth (SE)	11.1 (3.0)	5.6 (1.3)	−5.6 (3.1)	0.075
Extractions Performed on Deciduous Teeth	2.4 (1.0)	1.9 (0.8)	−0.5 (1.3)	0.716

SD = standard deviation; SE = standard error; * statistically significant *p* < 0.05.

## Data Availability

The data presented in this study are available on request from the corresponding author. The data are not publicly available due to the approved conditions stated in the ethics approval exemption principles.

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
