# Peer review of "A Cost Analysis of an Outreach School-Based Dental Program: Teeth on Wheels"

_children, 2021, doi:10.3390/children8020154_

Round 1

Reviewer 1 Report

The authors of the manuscript must make some modifications:

1) From line 59 to line 101 there is no bibliographic reference.

2) Few bibliographical references have been used in the introduction.

3) As this is an observational study, it would be useful to add the STROBE list in the manuscript.

4) How many examiners were involved in recording data in the study?

5) Why were restored permanent anterior teeth excluded?

Author Response

Reviewer 1

Comments to the Author

1) From line 59 to line 101 there is no bibliographic reference.

Response: The details presented here is via personal correspondence and not via any previous published work. We have added the reference [17] at Line 83:

‘17. (Abou-Zeid J.A. and Abou-Zeid R.A. 2020, personal communication, 7 December’

2) Few bibliographical references have been used in the introduction.

Response: We have added additional content, where appropriate, reflecting the background of school dental services and the literature around school-based dental services throughout the introduction. We have also considered the relevance of the oral health workforce in the revised paper.

3) As this is an observational study, it would be useful to add the STROBE list in the manuscript.

Response: Thankyou for this recommendation. We have included the STROBE checklist in this revised version.

4) How many examiners were involved in recording data in the study?

Response: The data source supplied does not include this information. We do not believe this is relevant since we have acknowledged there were methodological weaknesses in not using calibrated dental practitioners to measure dental caries experience. Data collected through dental service providers in general include multiple practitioners who are not calibrated.

5) Why were restored permanent anterior teeth excluded?

Response: Due to the nature of the data, the reasons for the restoration of anterior permanent teeth is unknown. It may be due to dental caries or dental trauma.  Therefore having an anterior tooth restored may not be due to the development of a new carious lesion but rather due to a traumatic incident which could not be prevented by the TOW program. Furthermore, posterior teeth are at a much greater risk of developing dental caries than anterior teeth.  We have revised this sentence at Line 57-61 to support this justification:

‘Due to the lack of data on the reasons for the restoration of permanent anterior teeth (which could be due to dental caries or trauma) and the fact that posterior permanent teeth are at a much greater risk of developing dental caries than anterior permanent teeth, it was decided to exclude restored anterior permanent teeth as a count for dental caries.’

Reviewer 2 Report

We read with great interest the manuscript with title “A Cost-Analysis of an Outreach School-Based Dental Program: Teeth On Wheels” aiming to assess on a sample of 414 children if a dental pro-active program called Teeth on Wheels would result in reduced costs and the number of restorations and extractions in the later latter period.

The topic is very interesting, even if at first reading it may seem like it has local relevance and is confined to certain regions of Australia. The text is well written, however some corrections are required.

Introduction

Line 97 “Additionally, educational toys and….”, please correct this sentence.

Author Response

Reviewer 2

Comments to the Author

We read with great interest the manuscript with title “A Cost-Analysis of an Outreach School-Based Dental Program: Teeth On Wheels” aiming to assess on a sample of 414 children if a dental pro-active program called Teeth on Wheels would result in reduced costs and the number of restorations and extractions in the later latter period.

The topic is very interesting, even if at first reading it may seem like it has local relevance and is confined to certain regions of Australia. The text is well written, however some corrections are required.

Response: Thankyou for your positive feedback. We have revised the paper to address any relevant grammar errors.

Reviewer 3 Report

Overall, there are some interesting findings in the study, taking into account that preventive dentistry has beneficial effect on reducing the incidence of decay. 

For future studies I recommend the use of calibrated examiners in order to provide reliable data.

The explanation of the studies limitations is appreciated and the findings should be interpreted with caution.

I suggest English minor spell check e.g.: line 18: “…later latter period.”, line 120, line 175, etc.

I recommend not to use so many self-citation (5 out 23). Please provide more available data by adding more references.

Author Response

Reviewer 3

Comments to the Author

Overall, there are some interesting findings in the study, taking into account that preventive dentistry has beneficial effect on reducing the incidence of decay. 

For future studies I recommend the use of calibrated examiners in order to provide reliable data.

Response: Thankyou for your positive feedback. We agree with the above key methodological weakness above and have further emphasised this here at Line 300-303:

‘Future research would be beneficial to determine the clinical effectiveness of children receiving dental care by the TOW program using more robust study methods to validate our study findings, particularly, the utilisation of calibrated dental practitioners to diagnose caries experience.’

The explanation of the studies limitations is appreciated and the findings should be interpreted with caution.

Response: Thankyou for your positive feedback.

I suggest English minor spell check e.g.: line 18: “…later latter period.”, line 120, line 175, etc.

Response: These corrections have been made.

I recommend not to use so many self-citation (5 out 23). Please provide more available data by adding more references.

Response: We have provided additional content in the introduction, with relevant references as well as a comparison of other studies in the discussion section. The reference list has now expanded 23 to 34. Several self-citations are important in the discussion piece since these studies are relevant to the population context.

Introduction

Line 97 “Additionally, educational toys and….”, please correct this sentence.

Response: We have corrected this to:

‘Additionally, educational toys are given as rewards…’

Round 2

Reviewer 1 Report

In material and methods, where have you described the STROBE checklist?

Author Response

Reviewer 1

Comments to the Author

In material and methods, where have you described the STROBE checklist?

Response: Thankyou for noting this missing detail. We have added the following sentence at Line 168-169'.

'The details of the study consistent with the STROBE checklist is provided as a supplementary material.'